# Gaussian Processes for Survival Analysis

**Tamara Fernández**
Department of Statistics,
University of Oxford.
Oxford, UK.
fernandez@stats.ox.ac.uk

**Nicolás Rivera**
Department of Informatics,
King's College London.
London, UK.
nicolas.rivera@kcl.ac.uk

**Yee Whye Teh**
Department of Statistics,
University of Oxford.
Oxford, UK.
y.w.teh@stats.ox.ac.uk

## Abstract

We introduce a semi-parametric Bayesian model for survival analysis. The model is centred on a parametric baseline hazard, and uses a Gaussian process to model variations away from it nonparametrically, as well as dependence on covariates. As opposed to many other methods in survival analysis, our framework does not impose unnecessary constraints in the hazard rate or in the survival function. Furthermore, our model handles left, right and interval censoring mechanisms common in survival analysis. We propose a MCMC algorithm to perform inference and an approximation scheme based on random Fourier features to make computations faster. We report experimental results on synthetic and real data, showing that our model performs better than competing models such as Cox proportional hazards, ANOVA-DDP and random survival forests.

## 1 Introduction

Survival analysis is a branch of statistics focused on the study of time-to-event data, usually called survival times. This type of data appears in a wide range of applications such as failure times in mechanical systems, death times of patients in a clinical trial or duration of unemployment in a population. One of the main objectives of survival analysis is the estimation of the so-called survival function and the hazard function. If a random variable has density function $f$ and cumulative distribution function $F$, then its survival function $S$ is $1 - F$, and its hazard $\lambda$ is $f/S$. While the survival function $S(t)$ gives us the probability a patient survives up to time $t$, the hazard function $\lambda(t)$ is the instant probability of death given that she has survived until $t$.

Due to the nature of the studies in survival analysis, the data contains several aspects that make inference and prediction hard. One important characteristic of survival data is the presence of many covariates. Another distinctive flavour of survival data is the presence of censoring. A survival time is censored when it is not fully observable but we have an upper or lower bound of it. For instance, this happens in clinical trials when a patient drops out the study.

There are many methods for modelling this type of data. Arguably, the most popular is the Kaplan-Meier estimator [13]. The Kaplan-Meier estimator is a very simple, nonparametric estimator of the survival function. It is very flexible and easy to compute, it handles censored times and requires no-prior knowledge of the nature of the data. Nevertheless, it cannot handle covariates naturally and no prior knowledge can be incorporated. A well-known method that incorporates covariates is the Cox proportional hazard model [3]. Although this method is very popular and useful in applications, a drawback of it, is that it imposes the strong assumption that the hazard curves are proportional and non-crossing, which is very unlikely for some data sets.

There is a vast literature of Bayesian nonparametric methods for survival analysis [9]. Some examples include the so-called Neutral-to-the-right priors [5], which models survival curves as $e^{-\tilde{\mu}((0,t])}$, where $\tilde{\mu}$ is a completely random measure on $\mathbb{R}^+$. Two common choices for $\tilde{\mu}$ are the Dirichlet process

[8] and the beta-Stacy process [20], the latter, being a bit more tractable due its conjugacy. Other alternatives place a prior on the hazard function, one example of this, is the extended gamma process [7]. The weakness of the above methods is that there is no natural nor direct way to incorporate covariates and thus, they have not been extensively used by practitioners of survival analysis. More recently, [4] developed a new model called ANOVA-DDP which mixes ideas from ANOVA and Dirichlet processes. This method successfully incorporates covariates without imposing strong constraints, though it is not clear how to incorporate expert knowledge. Within the context of Gaussian process, a few models has been considered, for instance [14] and [12]. Nevertheless these models fail to go beyond the proportional hazard assumption, which corresponds to one of the aims of this work. Another option is [11], which describes a survival model with non-proportional hazard and time-dependent covariates. Recently, we became aware of the work of [2], which uses a so-called accelerated failure times model. Here, the dependence of the failure times on covariates is modelled by rescaling time, with the rescaling factor modelled as a function of covariates with a Gaussian process prior. This model is different from our proposal, and is more complex to study and to work with.

Lastly, another well-known method is Random Survival Forest [10]. This can be seen as a generalisation of Kaplan Meier estimator to several covariates. It is fast and flexible, nevertheless it cannot incorporate expert knowledge and lacks interpretation which is fundamental for survival analysis.

In this paper we introduce a new semiparametric Bayesian model for survival analysis. Our model is able to handle censoring and covariates. Our approach models the hazard function as the multiplication of a parametric baseline hazard and a nonparametric part. The parametric part of our model allows the inclusion of expert knowledge and provides interpretability, while the nonparametric part allows us to handle covariates and to amend incorrect or incomplete prior knowledge. The nonparametric part is given by a non-negative function of a Gaussian process on $\mathbb{R}^+$.

Giving the hazard function $\lambda$ of a random variable $T$, we sample from it by simulating the first jump of a Poisson process with intensity $\lambda$. In our case, the intensity of the Poisson process is a function of a Gaussian process, obtaining what is called a Gaussian Cox process. One of the main difficulties of working with Gaussian Cox processes is the problem of learning the 'true' intensity given the data because, in general, it is impossible to sample the whole path of a Gaussian process. Nevertheless, exact inference was proved to be tractable by [1]. Indeed, the authors developed an algorithm by exploiting a nice trick which allows them to make inference without sampling the whole Gaussian process but just a finite number of points.

In this paper, we study basic properties of our prior. We also provide an inference algorithm based in a sampler proposed by [18] which is a refined version of the algorithm presented in [1]. To make the algorithm scale we introduce a random Fourier features to approximate the Gaussian process and we supply the respective inference algorithm. We demonstrate the performance of our method experimentally by using synthetic and real data.

## 2  Model

Consider a continuous random variable $T$ on $\mathbb{R}^+ = [0, \infty)$, with density function $f$ and cumulative distribution function $F$. Associated with $T$, we have the survival function $S = 1 - F$ and the hazard function $\lambda = f/S$. The survival function $S(t)$ gives us the probability a patient survives up to time $t$, while the hazard function $\lambda(t)$ gives us the instant risk of patient at time $t$.

We define a Gaussian process prior over the hazard function $\lambda$. In particular, we choose $\lambda(t) = \lambda_0(t)\sigma(l(t))$, where $\lambda_0(t)$ is a baseline hazard function, $l(t)$ is a centred stationary Gaussian process with covariance function $\kappa$, and $\sigma$ is a positive link function. For our implementation, we choose $\sigma$ as the sigmoidal function $\sigma = (1 + e^{-x})^{-1}$, which is a quite standard choice in applications. In this way, we generate $T$ as the first jump of the Poisson process with intensity $\lambda$, i.e. $T$ has distribution $\lambda(t)e^{-\int_0^t \lambda(s)ds}$. Our model for a data set of i.i.d. $T_i$, without covariates, is

$$l(\cdot) \sim \mathcal{GP}(0, \kappa), \qquad \lambda(t)|l, \lambda_0(t) = \lambda_0(t)\sigma(l(t)), \qquad T_i|\lambda \overset{iid}{\sim} \lambda(t)e^{-\int_0^{T_i} \lambda(s)ds}, \qquad (1)$$

which can be interpreted as a baseline hazard with a multiplicative nonparametric noise. This is an attractive feature as an expert may choose a particular hazard function and then the nonparametric noise amends an incomplete or incorrect prior knowledge. The incorporation of covariates is discussed later in this section, while censoring is discussed in section 3.

Notice that $\mathbf{E}(\sigma(X)) = 1/2$ for a zero-mean Gaussian random variable. Then, as we are working with a centred Gaussian process, it holds that $\mathbf{E}(\lambda(t)) = \lambda_0(t)\mathbf{E}(\sigma(l(t))) = \lambda_0(t)/2$. Hence, we can imagine our model as a random hazard centred in $\lambda_0(t)/2$ with a multiplicative noise. In the simplest scenario, we may take a constant baseline hazard $\lambda_0(t) = 2\Omega$ with $\Omega > 0$. In such case, we obtain a random hazard centred in $\Omega$, which is simply the hazard function of a exponential random variable with mean $1/\Omega$. Another choice might be $\lambda_0(t) = 2\beta t^{\alpha-1}$, which determines a random hazard function centred in $\beta t^{\alpha-1}$, which corresponds to the hazard function of the Weibull distribution, a popular default distribution in survival analysis.

In addition to the hierarchical model in (1), we include hyperparameters to the kernel $\kappa$ and to the baseline hazard $\lambda_0(t)$. In particular for the kernel, it is common to include a length scale parameter and an overall variance.

Finally, we need to ensure the model we proposed defines a well-defined survival function, i.e. $S(t) \to 0$ as $t$ tends to infinity. This is not trivial as our random survival function is generated by a Gaussian process. The next proposition, proved in the supplemental material, states that under suitable regularity conditions, the prior defines proper survival functions.

**Proposition 1.** *Let $(l(t))_{t \geq 0} \sim \mathcal{GP}(0, \kappa)$ be a stationary continuous Gaussian process. Suppose that $\kappa(s)$ is non-increasing and that $\lim_{s \to \infty} \kappa(s) = 0$. Moreover, assume it exists $K > 0$ and $\alpha > 0$ such that $\lambda_0(t) \geq Kt^{\alpha-1}$ for all $t \geq 1$. Let $S(t)$ be the random survival function associated with $(l(t))_{t \geq 0}$, then $\lim_{t \to \infty} S(t) = 0$ with probability 1.*

Note the above proposition is satisfied by the hazard functions of the Exponential and Weibull distributions.

## 2.1 Adding covariates

We model the relation between time and covariates by the kernel of the Gaussian process prior. A simple way to generate kernels in time and covariates is to construct kernels for each covariate and time, and then perform basic operation of them, e.g. addition or multiplication. Let $(t, X)$ denotes a time $t$ and with covariates $X \in \mathbb{R}^d$. Then for pairs $(t, X)$ and $(s, Y)$ we can construct kernels like

$$\hat{K}((t, X), (s, Y)) = \hat{K}_0(t, s) + \sum_{j=1}^{d} \hat{K}_j(X_j, Y_j),$$

or, the following kernel, which is the one we use in our experiments,

$$K((t, X), (s, Y)) = K_0(t, s) + \sum_{j=1}^{d} X_j Y_j K_j(t, s). \tag{2}$$

Observe that the first kernel establishes an additive relation between time and covariates while the second creates an interaction between the value of the covariates and time. More complicated structures that include more interaction between covariates can be considered. We refer to the work of [6] for details about the construction and interpretation of the operations between kernels. Observe the new kernel produces a Gaussian process from the space of time and covariates to the real line, i.e it has to be evaluated in a pair of time and covariates.

The new model to generate $T_i$, assuming we are given the covariates $X_i$, is

$$l(\cdot) \sim \mathcal{GP}(0, K), \quad \lambda_i(t)|l, \lambda_0(t), X_i = \lambda_0(t)\sigma(l(t, X_i)), \quad T_i|\lambda_i \overset{indep}{\sim} \lambda(T_i)e^{-\int_0^{T_i} \lambda_i(s)ds}, \tag{3}$$

In our construction of the kernel $K$, we choose all kernels $K_j$ as stationary kernels (e.g. squared exponential), so that $K$ is stationary with respect to time, so proposition 1 is valid for each fixed covariate $X$, i.e. giving a fix covariate $X$, we have $S_X(t) = \mathbf{P}(T > t|X) \to 0$ as $t \to \infty$.

## 3 Inference

### 3.1 Data augmentation scheme

Notice that the likelihood of the model in equation (3) has to deal with terms of the form $\lambda_i(t) \exp^{-\int_0^t \lambda_i(s)ds}$ as these expressions come from the density of the first jump of a non-homogeneous Poisson process with intensity $\lambda_i$. In general the integral is not analytically tractable since $\lambda_i$ is defined by a Gaussian process. A numerical scheme can be used, but it is approximate and

computationally expensive. Following [1] and [18], we develop a data augmentation scheme based on thinning a Poisson process that allows us to efficiently avoid a numerical method.

If we want to sample a time $T$ with covariate $X$, as given in equation (3), we can use the following generative process. Simulate a sequence of points $g_1, g_2, \ldots$ of points distributed according a Poisson process with intensity $\lambda_0(t)$. We assume the user is using a well-known parametric form, then sampling the points $g_1, g_2, \ldots$ is tractable (in the Weibull case this can be easily done). Starting from $k = 1$ we accept the point $g_k$ with probability $\sigma(l(g_k, X))$. If it is accepted we set $T = g_k$, otherwise we try the point $g_{k+1}$ and repeat. We denote by $G$ the set of rejected point, i.e. if we accepted $g_k$, then $G = \{g_1, \ldots, g_{k-1}\}$. Note the above sampling procedure needs to evaluate the Gaussian process in the points $(g_k, X)$ instead the whole space.

Following the above scheme to sample $T$, the following proposition can be shown.

**Proposition 2.** *Let* $\Lambda_0(t) = \int_0^T \lambda_0(t)dt$*, then*

$$p(G, T | \lambda_0, l(t)) = \left( \lambda_0(T) \prod_{g \in G} \lambda_0(g) \right) e^{-\Lambda_0(T)} \left( \sigma(l(T)) \prod_{g \in G} (1 - \sigma(l(g))) \right) \tag{4}$$

*Proof sketch.* Consider a Poisson process on $[0, \infty)$ with intensity $\lambda_0(t)$. Then, the first term in the RHS of equation (4) is the density of putting points exactly in $G \cup \{T\}$. The second term is the probability of putting no points in $[0, T] \setminus (G \cup \{T\})$, i.e. $e^{-\Lambda_0(T)}$. The second term is independent of the first one. The last term comes from the acceptance/rejection part of the process. The points $g \in G$ are rejected with probability $1 - \sigma(g)$, while the point $T$ is accepted with probability $\sigma(T)$. Since the acceptance/rejection of points is independent of the Poisson process we get the result. □

Using the above proposition, the model of equation (1) can be reformulated as the following tractable generative model:

$$l(\cdot) \sim \mathcal{GP}(0, K), \quad (G, T) | \lambda_0(t), l(t) \sim e^{-\Lambda_0(T)} (\sigma(l(T)) \lambda_0(T)) \prod_{g \in G} (1 - \sigma(l(g))) \lambda_0(g). \tag{5}$$

Our model states a joint distribution for the pair $(G, T)$ where $G$ is the set of rejected jump point of the thinned Poisson process and $T$ is the first accepted one.

To perform inference we need data $(G_i, T_i, X_i)$, whereas we only receive points $(T_i, X_i)$. Thus, we need to sample the missing data $G_i$ given $(T_i, X_i)$. The next proposition gives us a way to do this.

**Proposition 3.** *[18] Let* $T$ *be a data point with covariate* $X$ *and let* $G$ *be its set of rejected points. Then the distribution of* $G$ *given* $(T, X, \lambda_0, l)$ *is distributed as a non-homogeneous Poisson process with intensity* $\lambda_0(t)(1 - \sigma(l(t, X)))$ *on the interval* $[0, T]$.

### 3.2 Inference algorithm

The above data augmentation scheme suggests the following inference algorithm. For each data point $(T_i, X_i)$ sample $G_i | (T_i, X_i, \lambda_0, l)$, then sample $l | ((G_i, T_i, X_i)_{i=1}^n, \lambda_0)$, where $n$ is the number of data points. Observe that the sampling of $l$ given $(G_i, T_i, X_i)_{i=1}^n, \lambda_0)$ can be seen as a Gaussian process binary classification problem, where the points $G_i$ and $T_i$ represent two different classes. A variety of MCMC techniques can be used to sample $l$, see [15] for details.

For our algorithm we use the following notation. We denote the dataset as $(T_i, X_i)_{i=1}^n$. The set $G_i$ refers to the set of rejected points of $T_i$. We denote $\boldsymbol{G} = \bigcup_{i=1}^n G_i$ and $\boldsymbol{T} = \{T_1, \ldots, T_n\}$ for the whole set of rejected and accepted points, respectively. For a point $t \in G_i \cup \{T_i\}$ we denote $l(t)$ instead of $l(t, X_i)$, but remember that each point has an associated covariate. For a set of points $A$ we denote $l(A) = \{l(a) : a \in A\}$. Also $\Lambda_0(t)$ refers to $\int_0^t \lambda_0(s)ds$ and $\Lambda_0(t)^{-1}$ denotes its inverse function (it exists since $\Lambda_0(t)$ is increasing). Finally, $N$ denotes the number of iterations we are going to run our algorithm. The pseudo code of our algorithm is given in Algorithm 1.

Lines 2 to 11 sample the set of rejected points $G_i$ for each survival time $T_i$. Particularly lines 3 to 5 use the Mapping theorem, which tells us how to map a homogeneous Poisson process into a non-homogeneous with the appropriate intensity. Observe it makes uses of the function $\Lambda_0$ and its

**Algorithm 1:** Inference Algorithm.

---

**Input:** Set of times $\boldsymbol{T}$ and the Gaussian proces $l$ instantiated in $\boldsymbol{T}$ and other initial parameters

1  **for** *q=1:N* **do**
2     **for** *i=1:n* **do**
3        $n_i \sim \text{Poisson}(1; \Lambda_0(T_i))$;
4        $\tilde{C}_i \sim U(n_i; 0, \Lambda_0(T_i))$;
5        Set $A_i = \Lambda_0^{-1}(\tilde{A}_i)$;
6     Set $\boldsymbol{A} = \cup_{i=1}^{n} A_i$
7     Sample $l(\boldsymbol{A})|l(\boldsymbol{G} \cup \boldsymbol{T}), \lambda_0$
8     **for** *i=1:n* **do**
9        $U_i \sim U(n_i; 0, 1)$
10       set $G_{(i)} = \{a \in A_i$ such that $U_i < 1 - \sigma(l(a))\}$
11     Set $\boldsymbol{G} = \cup_{i=1}^{n} G_i$
12     **Update parameters of** $\lambda_0(t)$
13     **Update** $l(\boldsymbol{G} \cup \boldsymbol{T})$ and hyperparameter of the kernel.

---

inverse function, which shall be provided or be easily computable. The following lines classify the points drawn from the Poisson process with intensity $\lambda_0$ in the set $G_i$ as in proposition 3. Line 7 is used to sample the Gaussian process in the set of points $\boldsymbol{A}$ given the values in the current set $\boldsymbol{G} \cup \boldsymbol{T}$. Observe that at the beginning of the algorithm, we have $\boldsymbol{G} = \emptyset$.

### 3.3 Adding censoring

Usually, in Survival analysis, we encounter three types of censoring: right, left and interval censoring. We assume each data point $T_i$ is associated with an (observable) indicator $\delta_i$, denoting the type of censoring or if the time is not censored. We describe how the algorithm described before can easily handle any type of censorship.

**Right censorship:** In presence of right censoring, the likelihood for a survival time $T_i$ is $S(T_i)$. The related event in terms of the rejected points correspond to do not accept any location $[0, T_i]$. Hence, we can treat right censorship in the same way as the uncensored case, by just sampling from the distribution of the rejected jump times prior $T_i$. In this case, $T_i$ is not an accepted location, i.e. $T_i$ is not considered in the set $\boldsymbol{T}$ of line 7 nor 13.

**Left censorship:** In this set-up, we know the survival time is at most $T_i$, then the likelihood of such time is $F(T_i)$. Treating this type of censorship is slightly more difficult than the previous case because the event is more complex. We ask for accepting at least one jump time prior $T_i$, which might leads us to have a larger set of latent variables. In order to avoid this, we proceed by imputing the 'true' survival time $T_i'$ by using its truncated distribution on $[0, T_i]$. Then we proceed using $T_i'$ (uncensored) instead of $T_i$. We can sample $T_i'$ as following: we sample the first point of a Poisson process with the current intensity $\lambda$, if such point is after $T_i$ we reject the point and repeat the process until we get one. The imputation step has to be repeated at the beginning of each iteration.

**Interval censorship:** If we know that survival time lies in the interval $I = [S_i, T_i]$ we can deal with interval censoring in the same way as left censoring but imputing the survival time $T_i'$ in $I$.

## 4 Approximation scheme

As shown is algorithm 1, in line 7 we need to sample the Gaussian process $(l(t))_{t \geq 0}$ in the set of points $\boldsymbol{A}$ from its conditional distribution, while in line 13, we have to update $(l(t))_{t \geq 0}$ in the set $\boldsymbol{G} \cup \boldsymbol{T}$. Both lines require matrix inversion which scales badly for massive datasets or for data $\boldsymbol{T}$ that generates a large set $\boldsymbol{G}$. In order to help the inference we use a random feature approximation of the Kernel [17].

We exemplify the idea on the kernel we use in our experiment, which is given by $K((t, X), (s, Y)) = K_0(t, s) + \sum_{j=1}^{d} X_j Y_j K_j(t, s)$, where each $K_j$ is a square exponential kernel, with overall variance

$\sigma_j^2$ and length scale parameter $\phi_j$ Hence, for $m \geq 0$, the approximation of our Gaussian process is given by

$$g^m(t, X) = g_0^m(t) + \sum_{j=1}^d X_j g_j^m(t) \qquad (6)$$

where each $g_j^m(t) = \sum_{k=1}^m a_k^j \cos(s_k^j t) + b_k^j \sin(s_k^j t)$, and each $a_k^j$ and $b_k^j$ are independent samples of $\mathcal{N}(0, \sigma_j^2)$ where $\sigma_j^2$ is the overall variance of the kernel $K_j$. Moreover, $s_k^j$ are independent samples of $\mathcal{N}(0, 1/(2\pi\phi_j))$ where $\phi_j$ is the length scale parameter of the kernel $K_j$. Notice that $g^m(t, X)$ is a Gaussian process since each $g_j^m(t)$ is the sum of independent normally distributed random variables. It is know that as $m$ goes to infinity, the kernel of $g^m(t, X)$ approximates the kernel $K_j$. The above approximation can be done for any stationary kernel and we refer the reader to [17] for details.

The inference algorithm for this scheme is practically the same, except for two small changes. The values $l(A)$ in line 7 are easier to evaluate because we just need to know the values of the $a_k^j$ and $b_k^j$, and no matrix inversion is needed. In line 13 we just need to update all values $a_j^k$ and $b_j^k$. Since they are independent variables there is no need for matrix inversion.

## 5  Experiments

All the experiments are performed using our approximation scheme of equation (6) with a value of $m = 50$. Recall that for each Gaussian process, we used a squared exponential kernel with overall variance $\sigma_j^2$ and length scale parameter $\phi_j$. Hence for a set of $d$ covariates we have a set of $2(d+1)$ hyper-parameters associated to the Gaussian processes. In particular, we follow a Bayesian approach and place a log-Normal prior for the length scale parameter $\phi_j$, and a gamma prior (inverse gamma is also useful since it is conjugate) for the variance $\sigma_j^2$. We use elliptical slice sampler [16] for jointly updating the set of coefficients $\{a_k^j, b_k^j\}$ and length-scale parameters.

With respect the baseline hazard we consider two models. For the first option, we choose the baseline hazard $2\beta t^{\alpha-1}$ of a Weibull random variable. Following a Bayesian approach, we choose a gamma prior on $\beta$ and a uniform $U(0, 2.3)$ on $\alpha$. Notice the posterior distribution for $\beta$ is conjugate and thus we can easily sample from it. For $\alpha$, use a Metropolis step to sample from its posterior. Additionally, observe that for the prior distribution of $\alpha$, we constrain the support to $(0, 2.3)$. The reason for this is because the expected size of the set $\mathbf{G}$ increases with respect to $\alpha$ and thus slow down computations.

As second alternative is to choose the baseline hazard as $\lambda_0(t) = 2\Omega$, with gamma prior over the parameter $\Omega$. The posterior distribution of $\Omega$ is also gamma. We refer to both models as the Weibull model (W-SGP) and the Exponential model (E-SGP) respectively.

The implementation for both models is exactly the same as in Algorithm 1 and uses the same hyper-parameters described before. As the tuning of initial parameters can be hard, we use the maximum likelihood estimator as initial parameters of the model.

### 5.1  Synthetic Data

In this section we present experiments made with synthetic data. Here we perform the experiment proposed in [4] for crossing data. We simulate $n = 25, 50, 100$ and 150 points from each of the following densities, $p_0(t) = \mathcal{N}(3, 0.8^2)$ and $p_1(t) = 0.4\mathcal{N}(4, 1) + 0.6\mathcal{N}(2, 0.8^2)$, restricted to $\mathbb{R}^+$. The data contain the sample points and a covariate indicating if such points were sampled from the p.d.f $p_0$ or $p_1$. Additionally, to each data point, we add 3 noisy covariates taking random values in the interval $[0, 1]$. We report the estimations of the survival functions for the Weibull model in figure 1 while the results for the Exponential model are given in the supplemental material.

It is clear that for the clean data (without extra noisy covariates), the more data the better the estimation. In particular, the model perfectly detects the cross in the survival functions. For the noisy data we can see that with few data points the noise seems to have an effect in the precision of our estimation in both models. Nevertheless, the more points the more precise is our estimate for the survival curves. With 150 points, each group seems to be centred on the corresponding real survival function, independent of the noisy covariates.

We finally remark that for the W-SGP and E-SGP models, the prior of the hazards are centred in a Weibull and a Exponential hazard, respectively. Since the synthetic data does not come from those

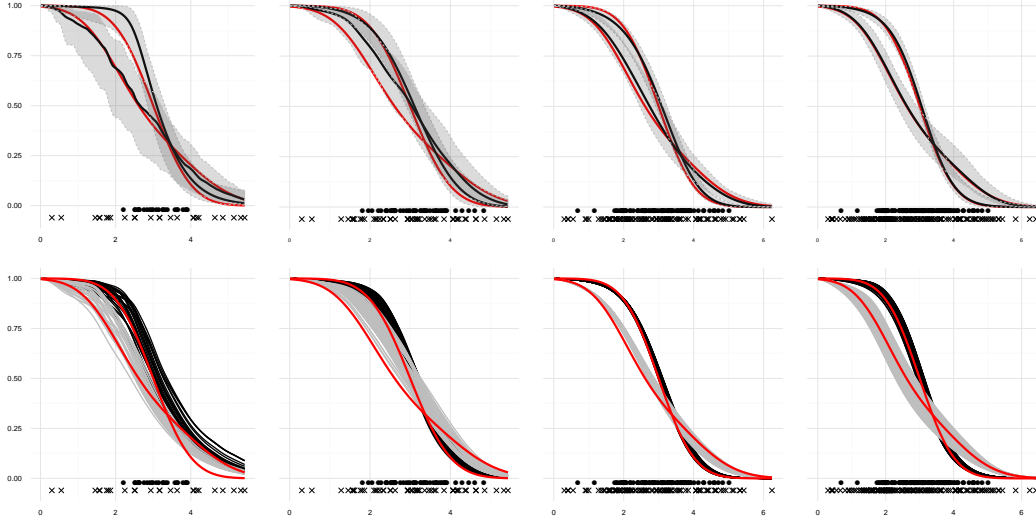

Figure 1: Weibull Model. First row: clean data, Second row: data with noise covariates. Per columns we have 25, 50, 100 and 150 data points per each group (shown in $X$-axis) and data is increasing from left to right. Dots indicate data is generated from $p_0$, crosses, from $p_1$. In the first row a credibility interval is shown. In the second row each curve for each combination of noisy covariate is given.

distributions, it will be harder to approximate the true survival function with few data. Indeed, we observe our models have problems at estimating the survival functions for times close to zero.

## 5.2 Real data experiments

To compare our models we use the so-called concordance index. The concordance index is a standard measure in survival analysis which estimates how good the model is at ranking survival times. We consider a set of survival times with their respective censoring indices and set of covariates $(T_1, \delta_1, X_1), \dots, (T_n, \delta_n, X_n)$. On this particular context, we just consider right censoring.

To compute the $C$-index, consider all possible pairs $(T_i, \delta_i, X_i; T_j, \delta_j, X_j)$ for $i \neq j$. We call a pair admissible if it can be ordered. If both survival times are right-censored i.e. $\delta_i = \delta_j = 0$ it is impossible to order them, we have the same problem if the smallest of the survival times in a pair is censored, i.e. $T_i < T_j$ and $\delta_i = 0$. All the other cases under this context will be called admissible. Given just covariates $X_i, X_j$ and the status $\delta_i, \delta_j$, the model has to predict if $T_i < T_j$ or the other way around. We compute the $C$-index by considering the number of pairs which were correctly sorted by the model, given the covariates, over the number of admissible pairs. A larger $C$-index indicates the model is better at predicting which patient dies first by observing the covariates. If the $C$-index close to $0.5$, it means the prediction made by the model is close to random.

We run experiments on the Veteran data, avaiable in the R-package survival package [19]. Veteran consists of a randomized trial of two treatment regimes for lung cancer. It has 137 samples and 5 covariates: **treatment** indicating the type of treatment of the patients, their **age**, the **Karnofsky performance score**, and indicator for **prior treatment** and **months from diagnosis**. It contains 9 censored times, corresponding to right censoring.

In the experiment we run our Weibull model (W-SGP) and Exponential model (E-SGP), ANOVA DDP, Cox Proportional Hazard and Random Survival Forest. We perform 10-fold cross validation and compute the $C$-index for each fold. Figure 2 reports the results.

For this dataset the only significant variable corresponds to the **Karnofsky performance score**. In particular as the values of this covariate increases, we expect an improved survival time. All the studied models achieve such behaviour and suggest a proportionality relation between the hazards. This is observable in the C-Index boxplot we can observe good results for proportional hazard rates.

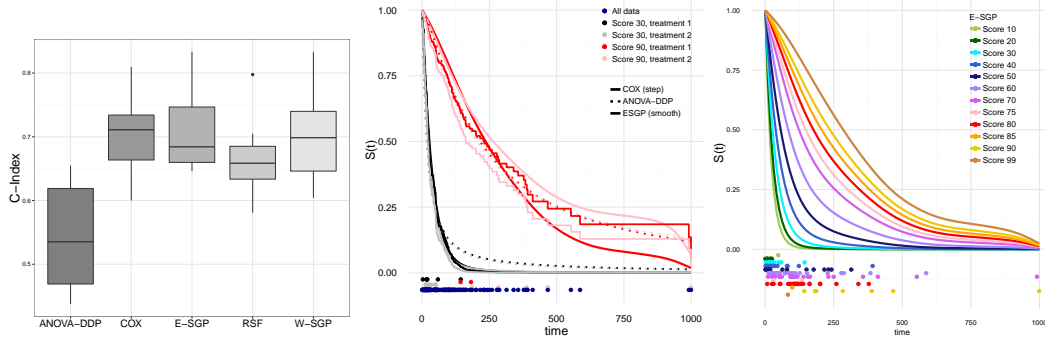

Figure 2: **Left**: C-Index for ANOVA-DDP,COX,E-SGP,RSF,W-SGP; **Middle**: Survival curves obtained for the combination of score: 30, 90 and treatments: 1 (standard) and 2 (test); **Right**: Survival curves, using W-SGP, across all scores for fixed treatment 1, diagnosis time 5 moths, age 38 and no prior therapy. (Best viewed in colour)

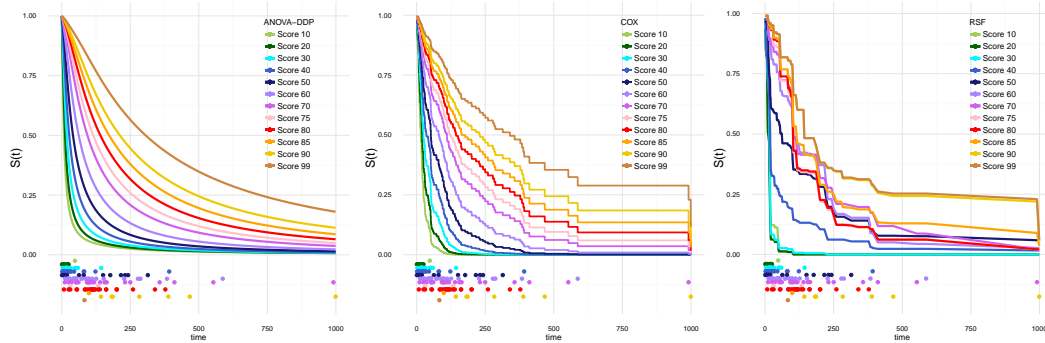

Figure 3: Survival curves across all scores for fixed treatment 1, diagnosis time 5 months, age 38 and no prior therapy. **Left:** ANOVA-DDP; **Middle:** Cox proportional; **Right:** Random survival forests.

Nevertheless, our method detect some differences between the treatments when the **Karnofsky performance score** is 90, as it can be seen in figure 2.

For the other competing models we observe an overall good result. In the case of ANOVA-DDP we observe the lowest C-INDEX. In figure 3 we see that ANOVA-DDP seems to be overestimating the Survival function for lower scores. Arguably, our survival curves are more visually pleasant than Cox proportional hazards and Random Survival Trees.

# 6 Discussion

We introduced a Bayesian semiparametric model for survival analysis. Our model is able to deal with censoring and covariates. In can incorporate a parametric part, in which an expert can incorporate his knowledge via the baseline hazard but, at the same time, the nonparametric part allows the model to be flexible. Future work consist in create a method to choose initial parameter to avoid sensitivity problems at the beginning. Construction of kernels that can be interpreted by an expert is something desirable as well. Finally, even though the random features approximation is a good approach and helped us to run our algorithm in large datasets, it is still not sufficient for datasets with a massive number of covariates, specially if we consider a large number of interactions between covariates.

### Acknowledgments

YWT's research leading to these results has received funding from the European Research Council under the European Union's Seventh Framework Programme (FP7/2007-2013) ERC grant agreement no. 617071. Tamara Fernández and Nicolás Rivera were supported by funding from Becas CHILE.

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
