[Supplementary Material]

# Supplemental Material for Gaussian Processes for Survival Analysis

**Tamara Fernández**
Department of Statistics,
University of Oxford.
Oxford, UK.
fernandez@stats.ox.ac.uk

**Nicolás Rivera**
Department of Informatics,
King's College London.
London, UK.
nicolas.rivera@kcl.ac.uk

**Yee Whye Teh**
Department of Statistics,
University of Oxford.
Oxford, UK.
y.w.teh@stats.ox.ac.uk

## 1   Proofs of propositions

**Proposition 1.** *Let $(l(t))_{t \geq 0} \sim \mathcal{GP}(0, \kappa)$ be a stationary continuous Gaussian process. Suppose that $\kappa(s)$ is non-increasing and that $\lim_{s \to \infty} \kappa(s) = 0$. Moreover, assume it exists $K > 0$ and $\alpha > 0$ such that $\lambda_0(t) \geq Kt^{\alpha-1}$ for all $t \geq 1$. Let $S(t)$ be the random survival function associated with $(l(t))_{t \geq 0}$, then $\lim_{t \to \infty} S(t) = 0$ with probability 1.*

*Proof.* Denote with $\mathbf{P}$ the probability associated with the gaussian process $(l(t))_{t \geq 0} \sim \mathcal{GP}(0, \kappa)$ and by $\mathbf{E}$ the corresponding expected values.

Remember our (random) hazard is given by $\lambda(s) = \lambda_0(s)\sigma(l(s)) \geq Ks^{\alpha-1}\sigma(l(s)) \geq 0$ for $s \geq 1$. It is well-known that the survival function can be written as $S(t) = e^{-\int_0^t \lambda(s)ds}$, then

$$S(t) = e^{-\int_0^t \lambda(s)ds} \leq e^{-\int_0^1 \lambda(s) - \int_1^t Ks^{\alpha-1}\sigma(l(s))ds} \leq e^{-\int_1^t Ks^{\alpha-1}\sigma(l(s))ds}$$

for $t \geq 1$.

We just need to prove that the latter term tends to 0 as $t$ goes to infinity. Consider the stochastic process $(X_t)_{t \geq 0}$ given by $X_t = \int_1^t Kt^{\alpha-1}\sigma(l(s))ds$. We compute the expected value and variance of $X_t$. By Tonelli's Theorem we have that

$$
\begin{aligned}
\mathbf{E}(X_t) &= K\mathbf{E}\left(\int_1^t s^{\alpha-1}\sigma(l(s))ds\right) \\
&= K\int_1^t s^{\alpha-1}\mathbf{E}(\sigma(l(s)))ds \\
&= \frac{K(t^{\alpha}-1)}{2\alpha}
\end{aligned}
\tag{1}
$$

In the last equality we used that $\mathbf{E}(\sigma(l(s))) = 1/2$ since the function $f(x) = \sigma(x) - 1/2$ is odd.

For the variance, we use Tonelli's Theorem, again, to obtain

$$
\begin{aligned}
\mathbf{Var}(X_t) &= K^2\mathbf{Var}\left(\int_1^t \sigma(l(s))x^{\alpha-1}ds\right) \\
&= K^2\int_1^t \int_1^t \mathbf{Cov}(\sigma(l(x)), \sigma(l(y)))(xy)^{\alpha-1}dxdy
\end{aligned}
\tag{2}
$$

We separate the last integral in two pieces, one integrating the region $A = \{t, s \in [1, t] : |t - s| < 1\}$ and its complement on $[1, t]^2$

In the region $A$ we use that $\mathbf{Cov}(\sigma(l(x)), \sigma(l(y))) \leq \sqrt{\mathbf{Var}(\sigma(l(x)))\mathbf{Var}(\sigma(l(y)))} = \mathbf{Var}(\sigma(l(0)))$ since $(l(t))_{t \geq 0}$ is stationary. Note that $\mathbf{Var}(\sigma(l(0))) \leq 1$ because $0 \leq \sigma(x) \leq 1$ for all $x$ Then

$$\int_A \mathbf{Cov}(\sigma(l(x)), \sigma(l(y)))(xy)^{\alpha-1}dxdy \quad \leq \quad \int_A (xy)^{\alpha-1}dxdy \tag{3}$$

a tedious computation gives us

$$\int_A \mathbf{Var}(\sigma(l(x)))(xy)^{\alpha-1}dxdy \leq \int_A (xy)^{\alpha-1}dxdy \leq C\frac{(t+1)^{2\alpha-1}}{2\alpha-1} \tag{4}$$

for some constant $C > 0$.

We claim the following inequality for all $(t, s) \in A^c$,

$$\mathbf{Cov}(\sigma(l(x)), \sigma(l(y))) \leq 2\frac{\kappa(|x-y|)\mathbf{E}(\sigma(l(x)))^2}{\kappa(0)-\kappa(1)}.$$

The proof of the above inequality is given in Lemma 1. Let $C > 0$ a large enough constant, then we have

$$\int_{A^c} \mathbf{Cov}(\sigma(l(x)), \sigma(l(y)))(xy)^{\alpha-1}dxdy \quad \leq \quad \int_{A^c} 2\frac{\kappa(|x-y|)\mathbf{E}(\sigma(l(x)))^2}{\kappa(0)-\kappa(1)}(xy)^{\alpha-1}dxdy$$
$$\leq \quad C\int_1^t \int_{x+1}^t \kappa(x-y)(xy)^{\alpha-1}dydx \tag{5}$$

Using the change of variables $w = x$ and $z = x - y$ we get from equation (5) that

$$\int_{A^c} \mathbf{Cov}(\sigma(l(x)), \sigma(l(y)))(xy)^{\alpha-1}dxdy \quad \leq \quad C\int_1^t \int_z^t \kappa(z)w^{\alpha-1}(w-z)^{\alpha-1}dwdz$$
$$\leq \quad C\int_1^t \int_1^t \kappa(z)w^{2\alpha-2}dwdz$$
$$\leq \quad C\frac{t^{2\alpha-1}}{2\alpha}\int_0^t \kappa(z)dz \tag{6}$$

Adding the integrals over $A$ and $A^c$, we get that it exists a large constant $C > 0$, depending on $\alpha$ such that for large enough $t$, it holds

$$\mathbf{Var}(X_t) \leq Ct^{2\alpha-1}\int_0^t \kappa(s)ds. \tag{7}$$

Then for large enough $t \geq 0$, by Chebyshev's inequality and equations (1) and (7) it holds

$$\mathbf{P}(|X_t - \mathbf{E}(X_t)| \geq \mathbf{E}(X_t)/2) \leq \frac{4\mathbf{Var}(X_t)}{\mathbf{E}(X_t)^2} = \mathcal{O}\left(\frac{t^{2\alpha-1}\int_0^t \kappa(s)ds}{t^{2\alpha}}\right) = \frac{o(t)}{t}. \tag{8}$$

In the last step we use that $\lim_{s \to \infty} \kappa(s) = 0$ which implies that $\int_0^t \kappa(s)ds = o(t)$.

Let $B_t$ be the event $B_t = \{|X_t - \mathbf{E}(X_t)| \geq \mathbf{E}(X_t)/2\}$. Let $(t_n)_{n \geq 1}$ be an increasing sequence of times, such that $\mathbf{P}(B_{t_n}) \leq n^{-2}$ and $t_n \to \infty$ as $n$ tends to $\infty$. Observe it is always possible to find such $t_n$ because equation (8). Observe $\sum_{n \geq 1} \mathbf{P}(B_{t_n}) \leq \infty$, then by using the Borel-Cantelli Lemma it holds that exists some finite $N \geq 1$ such that all event $B_{t_n}$ does not hold for $n \geq N$. Thus, for $n \geq N$ the equation

$$|X_{t_n} - \mathbf{E}(X_{t_n})| \leq \mathbf{E}(X_{t_n})/2,$$

holds true, implying that

$$X_{t_n} \geq \mathbf{E}(X_{t_n})/2.$$

Using the above equation, for $n \geq N$ we have

$$S(t_n) \leq e^{-X_{t_n}} \leq e^{-\mathbf{E}(X_{t_n})/2} \leq e^{-ct_n^\alpha},$$

for a small constant $c > 0$. Then since $S(t)$ is decreasing it holds

$$\lim_{t \to \infty} S(t) = \lim_{n \to \infty} S(t_n) \leq \lim_{n \to \infty} e^{-ct_n^\alpha} = 0.$$

$\square$

**Lemma 1.** *For any $t, s$ such that $|t - s| > 1$ we have*

$$\mathbf{Cov}(\sigma(l(x)), \sigma(l(y))) \leq 2 \frac{\kappa(|x - y|)\mathbf{E}(\sigma(l(x)))^2}{\kappa(0) - \kappa(1)}.$$

*Proof.* Let $|t - s| > 1$. Using that $xy \leq \frac{x^2 + y^2}{2}$ we have

$$
\begin{aligned}
\mathbf{E}(\sigma(l(t))\sigma(l(s))) \ &= \ \int_{-\infty}^{\infty} \int_{-\infty}^{\infty} \sigma(x)\sigma(y) \frac{\exp\left\{-\frac{\kappa(0)(x^2+y^2)-2\kappa(t-s)xy}{2(\kappa(0)^2-\kappa(t-s)^2)}\right\}}{2\pi(\kappa(0)^2 - \kappa(t - s)^2)^{1/2}} dxdy \\
&\leq \ \int_{-\infty}^{\infty} \int_{-\infty}^{\infty} \sigma(x)\sigma(y) \frac{\exp\left\{-\frac{(\kappa(0)-\kappa(t-s))(x^2+y^2)}{2(\kappa(0)^2-\kappa(t-s)^2)}\right\}}{2\pi(\kappa(0)^2 - \kappa(t - s)^2)^{1/2}} dxdy \\
&\leq \ \int_{-\infty}^{\infty} \int_{-\infty}^{\infty} \sigma(x)\sigma(y) \frac{\exp\left\{-\frac{(x^2+y^2)}{2(\kappa(0)+\kappa(t-s))}\right\}}{2\pi(\kappa(0)^2 - \kappa(t - s)^2)^{1/2}} dxdy \\
&\leq \ \frac{\kappa(0) + \kappa(t - s)}{\kappa(0) - \kappa(t - s)} \mathbb{E}(\sigma(l))^2 \leq \frac{\kappa(0) + \kappa(t - s)}{\kappa(0) - \kappa(1)} \mathbb{E}(\sigma(l))^2.
\end{aligned}
$$

In the last inequality we use that $k(s)$ is non-increasing. Finally, by deleting $\mathbf{E}(\sigma(l(0)))^2$ in both sides of the above equation gives us the covariance of $\sigma(l(t))$ and $\sigma(l(s))$, which give us the corresponding bound. $\qquad\square$

## 2 Survival Function for E-SGP

Figure 1: Exponential Model. First row: clean data, Second row: data with noisy covariates. Per columns we have 25,50,100 and 150 data points per each group (shown in $X$-axis) and data is increasing from left to right. Dots indicate data is generated from density $p_0$, crosses, from $p_1$. In the first row a confidence interval for each curve is given. In the second row each curve for each combination of noisy covariate is shown.