[Reviews · NeurIPS 2016]

Reviewer 1

Summary

The paper proposes a Gaussian process survival model. The idea is that a parametric survival model with form \lambda_0(t) is multiplied by a sigmoid-transformed GP, so \lambda(t) = \lambda_0(t)\sigma(l(t), l~GP. This formulation lets the authors incorporate covariates through the Gaussian process (they propose a suitable kernel in eq (2)).

Qualitative Assessment

A neat inference method is proposed based on thinning, since the survival model can be seen as the first-hitting-time of a Poisson process. I really liked this connection! It allows the paper to use the schemes proposed by [1], [13] to do MCMC. This is the coolest bit of the paper. I did have to sit with this for a while to understand why Proposition 2 is valid. Given that the paper is a bit short, it might have made sense to use some extra space to explain this concept more clearly. Using Fourier features seems like a sensible idea. It would have been interesting to hear how well you felt they approximated the GP (perhaps by comparing to a full GP on a simulated dataset?) I've heard, only annecdotally, that Fourier features are difficult to get to work with complex models like this, so I'd appreciate more detail on this. Although the paper reads beautifully for the first 5 pages, it becomes a little less clear in the experiments section. For example, It's not clear to me what the crosses and dots under the plots are in figure 2. I guess they are simulated data: but what's the difference between the dots and crosses? The ticks on Figs 2 and 3 are to small to read. It's not immediately clear in fig 2 that the plots have increasing data from left to right. Notes -- line 179 "can be done for any kernel". I'm not sure about that. Stationary kernels, yes. Polynomial kernels, yes. In general, I think no. I suggest you remove this statement unless you're absolutely certain that I'm wrong. line 169 "scales very bad"-> badly Equation 3: do you mean \lambda(T_i)e^{-\int ...} ? line 114. I think you mean Poisson process with intensity \lambda_0, not Gaussian process.

Confidence in this Review

2-Confident (read it all; understood it all reasonably well)


Reviewer 2

Summary

The paper considers survival analysis model where Gaussian processes (GP) are used to model the proportional hazard function. Authors propose also a new MCMC scheme to infer the model and perform a series of experiments where they compare their model to ANOVA-DDP and random survival forest

Qualitative Assessment

The paper contains some nice ideas concerning the inference of the proposed model and could potentially make a reasonable paper if rewritten. However, this requires more work than just revision. Here, I concentrate on the most important points. The introduction does not cover the background literature comprehensively. Gaussian processes have been proposed to survival analysis in the context of Cox proportional hazard model by at least Martino et al. (2011) and Joensuu et al. (2012) and the implementations for these models can be found from INLA and GPstuff packages. These work should at least be cited. Moreover, I think authors model should be compared to the existing GP approaches in the experiments (see below) The presented model differs from the earlier GP approaches to survival modeling essentially by choice of the link function and the baseline hazard (which Joensuu et al. (2012), for example, give a GP prior as well). The kernel is only a prior specification of the model and hence, Hence, the experiments should compare Authors . However, there are too many shortcuts taken or things left out of consideration that Authors make unconventional choice for the link function of the proportional hazard term by using sigmoidal link. Sara Martino, Rupali Akerkar, and Håvard Rue. Approximate Bayesian inference for survival models. Scandinavian Journal of Statistics, 38(3):514–528, 2011. Heikki Joensuu, Aki Vehtari, Jaakko Riihimäki, Toshirou Nishida, Sonja E Steigen, Peter Brabec, Lukas Plank, Bengt Nilsson, Claudia Cirilli, Chiara Braconi, Andrea Bordoni, Magnus K Magnusson, Zdenek Linke, Jozef Sufliarsky, Federico Massimo, Jon G Jonasson, Angelo Paolo Dei Tos, and Piotr Rutkowski. Risk of gastrointestinal stromal tumour recurrence after surgery: an analysis of pooled population-based cohorts. The Lancet Oncology, 13(3):265–274, 2012.

Confidence in this Review

3-Expert (read the paper in detail, know the area, quite certain of my opinion)


Reviewer 3

Summary

This paper suggest a semi-parametric model for survival analysis. The approach is based on modelling the hazard function which is the density divided by the complement to the cumulative distribution. The hazard function is modelled using a product of a parameteric function and a non-parametric function inside a non-linear link-function. In specific a gaussian process prior is placed over the non-parametric part of the function. Two different gaussian process priors are suggested in the paper, one addative covariance and one multiplicative. There are two sets of covariates, temporal and something else which in the paper is referred abstractly as "covariates". The first covariance is a sum of two different covariance functions while the other is a sum of a temporal covariance with product of the covariate with a different temporal covariance. The paper describes an approach to approximate the final covariance using a spectral decomposition which allows using larger data-sets. The paper describes an algorithmical approach to perform inference and shows experiments on both synthetic and real data.

Qualitative Assessment

I enjoyed reading this paper, not being terribly familiar with work on survival analysis but very comfortable with GPs it was a bit of a struggle for me to understand certain parts of the paper initially and there still might be things that I've misunderstood and not being completely familiar with the background material my novelty rating should be taken with a grain of salt. Initially I was confused about the model as it seemed very specific but not particularly well motivated. After a bit or reading up on related material I understood that this is very related to Gaussian Cox process, a reference would clarify things here or making the connection in the introductions two last paragraphs clearer between "our prior" and the Gaussian Cox process. The paper is clearly written and well but I wish the authors would make their contributions clearer without the reader having to resort to go through all the related material. This is especially true for Section 3 and the data augumentation. What is the motivation for the specific co-variances that you propose? There are many ways to combine multiple covariances, what makes the two proposed sensible? What is the actual form of K_0 and K_j that are used in the experiments? The only reference of hyper-parameters I find relates to them having a "length scale" but it doesn't say what the actual form is. In the algorithm section it seems that you need to have the hyper-paramters of the GP as input, how are they choosen? The proofs of the propositions are provided as supplementary material which is fine however the way they are presented in the paper they come a little bit abrupt. My suggestion is that as you still have about 1/2 page left to use that you provide a little bit of a sketch in the actual paper. There is a lot of spelling mistakes in the paper and some odd sentences. I've highlighted a few of them below but I recommend that the authors go through the paper in more detail. l35: The = There l52: gives us interpretation = provides interpretability l53: export = expert l62: After that we = We l79: Hence, he = Hence, we l84: In addition to = To l84: model of (1) = model in (1) l198: made in = made on l199: survival function = survival functions Caption Figure 2: courve = curve

Confidence in this Review

1-Less confident (might not have understood significant parts)


Reviewer 4

Summary

Authors propose a model for reconstruction of survival function based on Gaussian processes. A the model is intractable they use MCMC to make a final inference. Provided experiments' results suggest that this model can outperform common survival analysis model in some cases.

Qualitative Assessment

I consider weak accept of the presented paper due to the following reasons. To my knowledge there is no previous applications of Gaussian process approach in survival analysis, so pioneering in this area is benefitial from both survival analysis and Gaussian processes communities. The provided approach is quite straigtforward combination of existed approaches, while in some cases it is a tricky problem to provide a Gaussian process kind model for some problems. However, approach proposed by authors has many parameters, so I expect that it will be hard to implement it in a general way. Authors suggest to use MCMC or other kind of sampling for some stages of their procedure. This decision significanlty increases computational expenses and has other drawbacks. It would be good to use some approximate inference approach like variational inference instead.

Confidence in this Review

1-Less confident (might not have understood significant parts)


Reviewer 5

Summary

The paper proposed a semi-parametric Bayesian model for survival analysis, where the baseline hazard is parametric and the variation is nonparametric given by a Gaussian process with covariate effects. Inference and approximation algorithms are developed and the performances are demonstrated using both simulation and real data application. In the real data application, performances are compared with existing methods such as Cox proportional hazards, ANOVA-DDP and random survival forests.

Qualitative Assessment

I think the paper is very well written and easy to follow. My major concern is the novelty and authors' arguments on the advantages of the proposed method. Novelty. With a rich literature on Gaussian processes, or more generally nonparametric (Bayes) approaches, it is not surprising that Gaussian processes can be applied to survival analysis without major modifications, from a nonparametric modeling of the hazard function to an incorporation of additional covariates. Therefore, I think it critical to emphasize the novelty of the paper, which may deserve more discussion and presentation than the current version. Advantages of the proposed method. Several existing approaches, as argued by the authors, fail to "incorporate expert knowledge" and sometimes "lack interpretation which is fundamental for survival analysis". However, from the simulations to the real data application, does the proposed method incorporate any expert knowledge or improve the interpretation? On a related note, this also helps to clarify the novelty, therefore, I think to make this argument more convincing and explicit seems to be important here. A minor concern is related to the experiment using synthetic data. Only one discrete covariate is considered in the setting, which makes the model a mixture of two possibilities. Continuous covariates seem to be more challenging. Since the real data application has both continuous and discrete covariates, it seems more consistent and also convincing if the synthetic data experiment does this too.

Confidence in this Review

3-Expert (read the paper in detail, know the area, quite certain of my opinion)


Reviewer 6

Summary

In the paper 'Gaussian Processes for Survival Analysis', the authors present a semi-parametric approach to survival time modeling in the presents of covariates. The proposed method combines expert-knowledge in form of the parametric component, and a Gaussian Process (GP) as a non-parametric part to model differences between the former and the data. The model, dubbed SGP, is compared to other, standard approaches in this area on synthetic and real world data. The paper thoroughly introduces the method in section 2 by walking the reader through the layers of the hierarchical model in Equation (1) which is later extended in Equation (3) to account for covariates. In essence, the survival times are modeled as the first jumps of a Poisson process with an intensity assumed to be the product of an hand-crafted base function times a transformed GP. The non-linear transformation of the GP assures that the model yields a proper survival function. Section 3 shows that the SGP is tractable by defining a generative process to sample a survival time needed to perform inference. Also the (approximate) treatment of censored data points is discussed. To be able to handle large data sets, Section 4 describes an approximation scheme to the kernel based on a random Fourier series. Section 5 presents the experiments on one synthetic benchmark and a real world example, namely the Veteran data set available in R. For the latter, SGP is compared to three other methods for survival analysis.

Qualitative Assessment

The general idea of the paper seems promising, but I have several concerns: 1. The kernel choices in Equation (2) are not motivated and quite surprising to me. While a base kernel in 't' only makes a lot of sense, I don't understand the additive part for the covariates. Especially for the one used in the experiments, the summation of the product the covariates X_j, Y_j, and a kernel K_j(t,s) seems counterintuitive. Why should the covariance between to points grow if one of the covariates increases? Why not use a ARD (automatic relevance detection) kernel for all covariates combined rather than summing over all of them independently? 2. There is a discussion about the noise-parameters and the length-scales of the kernel, but what I missed was the actual method to adjust them given the data. A maximum likelihood estimation is mentioned, but only for the initialization. From experience, I know that this is crucial for 'high quality' GPs. 3. The experiments section is rather week. There is no comparison to the other algorithms on the synthetic benchmark, and the size of the real world benchmark with 137 samples and 5 covariates seems to small for the effort spent on approximations earlier in the paper. This amount of data should be no problem for a proper GP treatment without any approximations to the kernel. 4. Using GPs together with a parametric function to incorporate expert knowledge is a good idea, but the presented method makes a few ad-hoc choices to combine these two, e.g. the positive link function, or the particular kernel choice. An alternative could be a non-zero mean GP, where the mean function incorporates the expert knowledge. By modeling, for example, log(lambda) with this method, one could potentially enforce positiveness of the rate while also ensuring that the Survival function is well defined. Overall, I think the paper is trying to accomplish too much at the same time: the GP based semi-parametric model, the (approximate) handling of censored data, the fast inference algorithm, and the random fourier kernel approximation. To me it feels that maybe just the first two point above without a focus the scalability, but with a more thorough empirical evaluation would make a more concise paper. The field of approximate GPs offers several possibilities to extend to larger data sets and many covariates. Besides the technical points, the presentation of the material requires improvement: there are a few typos; citations are frequently used as nouns; and sometimes certain paragraphs feel out of place. For example, in line 55, Gaussian Cox Processes are mentioned for the first (and I think last time) without any context, citation or further explanation. At this point, I have to recommend rejecting the paper, but I would like to encourage the authors to improve the paper for a future submission.

Confidence in this Review

2-Confident (read it all; understood it all reasonably well)